# Inverse-design magnonic devices

Qi Wang [1✉], Andrii V. Chumak [1] & Philipp Pirro [2]

The field of magnonics offers a new type of low-power information processing, in which magnons, the quanta of spin waves, carry and process data instead of electrons. Many magnonic devices were demonstrated recently, but the development of each of them requires specialized investigations and, usually, one device design is suitable for one function only. Here, we introduce the method of inverse-design magnonics, in which any functionality can be specified first, and a feedback-based computational algorithm is used to obtain the device design. We validate this method using the means of micromagnetic simulations. Our proof-of-concept prototype is based on a rectangular ferromagnetic area that can be patterned using square-shaped voids. To demonstrate the universality of this approach, we explore linear, nonlinear and nonreciprocal magnonic functionalities and use the same algorithm to create a magnonic (de-)multiplexer, a nonlinear switch and a circulator. Thus, inverse-design magnonics can be used to develop highly efficient rf applications as well as Boolean and neuromorphic computing building blocks.

[1] Faculty of Physics, University of Vienna, Vienna, Austria. [2] Fachbereich Physik and Landesforschungszentrum OPTIMAS, Technische Universität Kaiserslautern, Kaiserslautern, Germany. ✉email: qi.wang@univie.ac.at

Magnonics is an emerging field of science in which magnons, the quanta of spin waves, are used to carry and process information instead of electrons[1–4]. Spin waves are propagating disturbances in the spin system of a solid body that can be used for a low-loss information transfer without any motion of real particles[5–7]. They have wavelengths ranging from micrometers down to atomic scales[7–9] and show a variety of pronounced nonlinear phenomena[10–12] that makes spin waves to be promising for Boolean computing[13–15], radio frequency applications[16,17] and neuromorphic computing[18,19] at the nanoscale. Many magnonic devices have been demonstrated recently: frequency (de-)multiplexers[16,17], nonlinear units[12,19], nonreciprocal diodes/circulators[20–22], and an integrated magnonic half-adder[23]. However, the development of each of these devices requires specialized efforts-demanding and complex investigations. In contrast, modern electronics are much more advanced and the design of even the most complex circuits is usually performed using automatized electronic software such as VHDL[24]. Moreover, recently the field photonics, which operates with light waves to process data, proposed the approach of inverse-design devices[25–28]. The main idea of this approach is to create a special medium divided into small elements arranged in a matrix, in which the refractive index can be changed locally for every element. This medium can perform, in principle, any functionality by the proper tuning of the state of each element in the matrix. A feedback loop-based numerical optimization algorithm is used to define the required states of the matrix elements automatically. Different linear optical devices have been proposed and realized using this method[25–28]. However, nonlinear phenomena, e.g., the dependence of the wavelength on the amplitude, are strongly needed for a computing system, which is still a challenge due to the rather low intrinsic photonic nonlinearity[29]. Besides, (1) the pronounced natural nonlinearity, the field of magnonics offers (2) a scalability of the devices down to the scale of a few nanometers[6,30] and, thus the integrability with CMOS devices[13] and (3) pronounced nonreciprocal properties associated with the gyrotropic magnetization precession motion and breaking of time symmetry[20–22,31].

Here, we introduce the inverse-design method into the field of magnonics and demonstrate its high performance, flexibility, and potential. We show numerically that a design region of only $1 \times 1 \, \mu m^2$ is sufficient to perform conceptually different functionalities. To demonstrate this, we explore linear, nonlinear, and nonreciprocal functionalities of the device and use the same automatic feedback algorithm to develop a nanoscale magnonic (de-)multiplexer, a nonlinear switch and a circulator. The size of each element in the used $10 \times 10$ (or $10 \times 20$) matrix was chosen to be $100 \times 100 \, nm^2$, in order to ensure simplicity in the experimental realization of the matrix[6,30]. In addition, we show that this size can be easily decreased to, e.g., $10 \times 10 \, nm^2$ (so, the whole matrix is $100 \times 100 \, nm^2$) since the fundamental limitation of the element size is given by the lattice constant of a magnetic material (see the Supplementary Information). We are confident that inverse-design magnonics will be intensively used in future because it allows for the automatic (efforts-less) design of practically any spin-wave based rf device, any spin-wave logic gate for binary data processing as well as for more complex units of unconventional data processing like neuromorphic computing. It is noted, that our manuscript is submitted simultaneously with Ref. [32] in which the authors use the inverse-design magnonics approach to demonstrate the concept of a nanoscale neural network.

## Results

As the first example, the structure of the inverse-designed magnonic frequency demultiplexer is shown in Fig. 1. It is designed based on a 100-nm-thick Yttrium Iron Garnet (YIG) film[33] and consists of one input waveguide, two output waveguides with the same width of 300 nm, and a design region of $1 \times 1 \, \mu m^2$ between them. The design region has been divided into $10 \times 10$ elements, each with a size of $100 \times 100 \, nm^2$ in which the magnetic material (YIG) is allowed to be entirely removed. This structure can be fabricated using focused ion beam[34,35] or by $Ar^+$ ion etching[6,30,36,37] and further miniaturization is possible. The yellow regions in Fig. 1 indicate the YIG and the square-shaped white holes imply the empty areas. It is noted that here we just demonstrate the concept and, for simplicity, use a "binary approach" where each region can have two states: 1 (YIG) or 0 (vacuum). But since the complexity of the functionality of an inverse-design device is fundamentally limited by the number of the degrees of freedom[24,25], namely by the design space, the following approaches can be used: (1) The switch from the 2D to 3D matrices[38,39] or the incremental engineering of (2) the thickness of the elements[35,40], (3) the saturation magnetization[40–42], or (4) the exchange stiffness of the magnetic material[42]. An external field of 200 mT is applied out-of-plane along the $z$-axis, and Forward Volume Spin Waves (FVSWs) are investigated (please note that this is not a classical FV magnetostatic wave (FVMSW) in a plane film since here also exchange energy is taken into account). Spin waves of two frequencies $f_1 = 2.6 \, GHz$ (corresponds to a wavelength $\lambda = 2 \, \mu m$) and $f_2 = 2.8 \, GHz$ ($\lambda = 1 \, \mu m$) are excited in the input waveguide and are separated into the different output waveguides after they pass through the design region optimized by a modified version of the direct binary search (DBS) algorithm[28,43].

The DBS algorithm is an optimization method that sequentially evaluates every element of the design region. The optimization process of the DBS algorithm is shown in Fig. 2a. First, a random initial saturation magnetization distribution matrix $M_0(m,n)$ ($m \in [1, 10], n \in [1, 10]$ are the indices of the element) is created in the design region, in which the material of a few elements has been randomly removed as shown in Fig. 2(b). Subsequently, the initial saturation magnetization distribution is introduced into Mumax[3], a micromagnetic simulation software[44]. The ground state is obtained by relaxation and subsequently, the spin-wave propagation in the structure is simulated to calculate the spin-wave intensity in the output waveguides. The excitation and detection regions are marked by red regions in Fig. 2b. The details of the micromagnetic simulations can be found in Methods. The objective function, which is to be maximized by the DBS algorithm, is a crucial control knob in the design and varies depending on the different optimization goals. For the magnonic demultiplexer, we define the objective function as:

$$O = (I_{f_1, o_1} - I_{f_1, o_2})(I_{f_2, o_2} - I_{f_2, o_1}) \quad (1)$$

where $I_{f_i, o_j}$ is the spin-wave intensity of frequency $f_i$ at the $j$-th output waveguide ($i, j = 1, 2$). The values of each bracket are evaluated individually and only positive values are accepted. The aim of this objective function is to maximize the transmission of the spin wave of frequency $f_1$ ($f_2$) to the output waveguide 1 (2) and simultaneously, to minimize the crosstalk between them, in short, i.e., to realize the function of the demultiplexer.

Second, a random indices combination $(m, n)$ is produced. If the magnetization of the initial element $M_0(m,n)$ is zero, i.e., the material has been removed, the magnetization of this element is toggled to one, i.e., filled with YIG, or vice versa as indicated in the flow chart of Fig. 2a. The updated magnetization distribution is loaded into Mumax[3] again to calculate a new objective value denoted as $O_{m,n}$. If this new one is better than the previous one, i.e., $O_{m,n} > O$, then the change of the magnetization of the element is accepted, and then the value of $O$ is replaced by the new one $O_{m,n}$ for the next evaluation. Otherwise, the change of the magnetization is undone. This procedure is repeated until all the elements have been evaluated, and then the magnetization

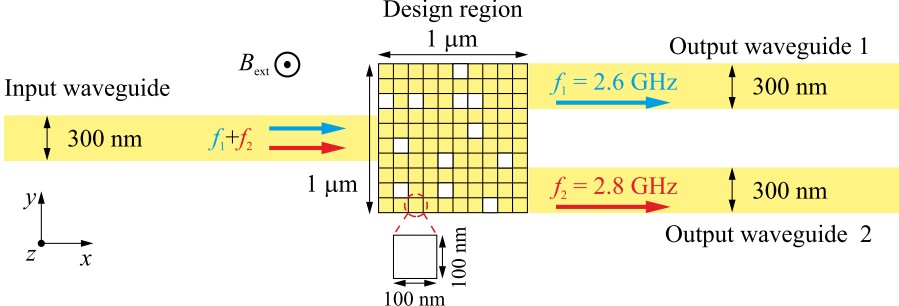

**Fig. 1 Structure of the inverse-designed magnonic frequency demultiplexer.** The magnonic frequency multiplexer consists of one input waveguide and two output waveguides with a width of 300 nm which are connected by a $1 \times 1\,\mu m^2$ design region. The design region has been divided into $10 \times 10$ elements each with a size of $100 \times 100\,nm^2$. The yellow region indicates the magnetic material (YIG) and the white holes imply that the material has been fully removed. An external field of 200 mT is applied along the $z$-axis.

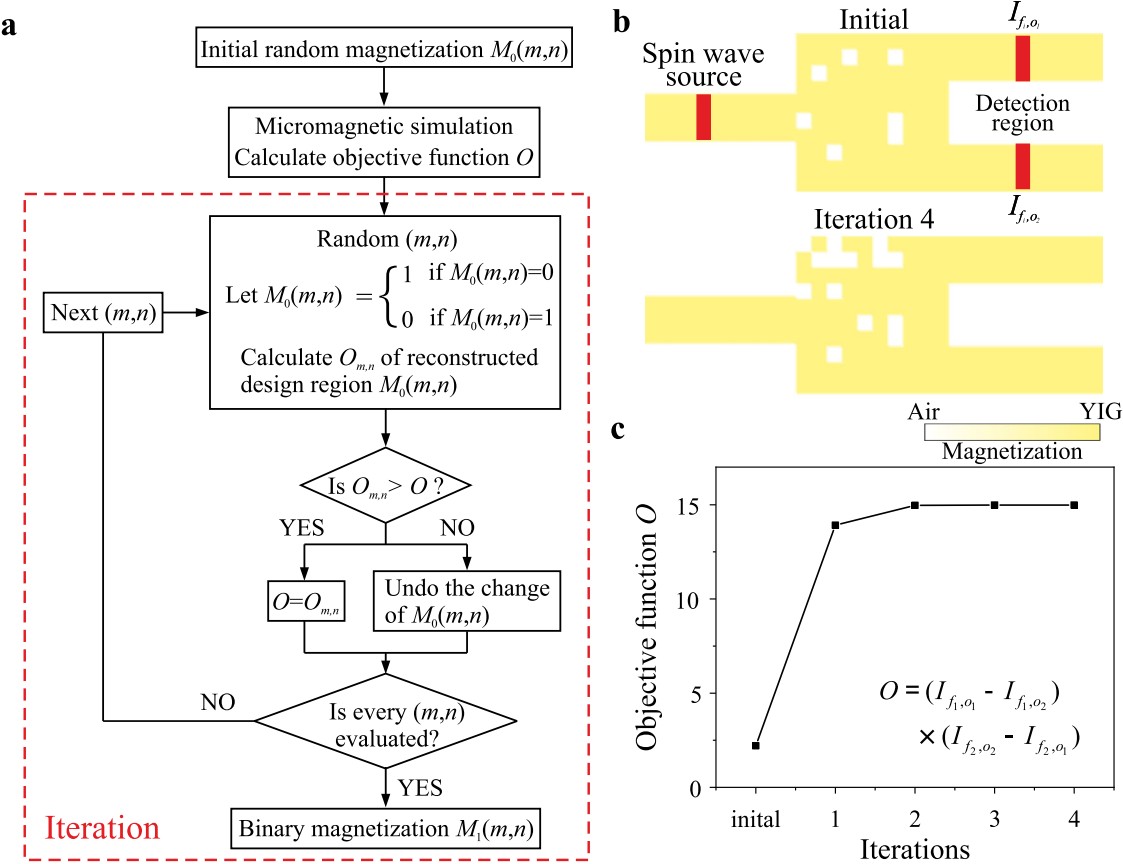

**Fig. 2 The optimization procedure of direct binary search algorithm. a** Flow chart of the Direct Binary Search (DBS) algorithm for one iteration. Every iteration includes 100 simulations in our examples. **b** The random initial magnetization distribution and the magnetization distribution after four iterations. The red regions show the excitation and detection regions. **c** Evolution of objective function $O$ as a function of the iteration number. The DBS algorithm has shown a particularly high efficiency in the case of spin waves and all the designs reported further required below five iterations only.

distribution after this first iteration is marked as $M_1(m,n)$ and will be used as an initial state for the second round. The whole optimization process finishes when the objective value exhibits no further improvement after one iteration or the improvement is smaller than a threshold (<1% for our case). The evolution of the objective function $O$ is plotted in Fig. 2c as a function of the iteration number, which converges after three iterations. The fast speed of convergence is due to the randomized selection order of elements rather than the sequential selection order of elements in the traditional DBS algorithm[45]. For our case, the 100 elements in the design region would provide $2^{100}$ element combinations and

it is obviously not possible to evaluate all the possibilities. One can expect that there should be a number of possible element combinations which show similar performances comparing with the global maximum. However, the DBS algorithm is sensitive to the initial state and also to the selection order of the elements and tends to suffer from premature convergence in local maxima. Therefore, we repeat the same optimization procedure several times with random initial conditions and chose the best design with the largest objective function as shown in Fig. 2b (bottom panel). Therefore, although the potential of the inverse-design magnonics is shown already here, only the further development

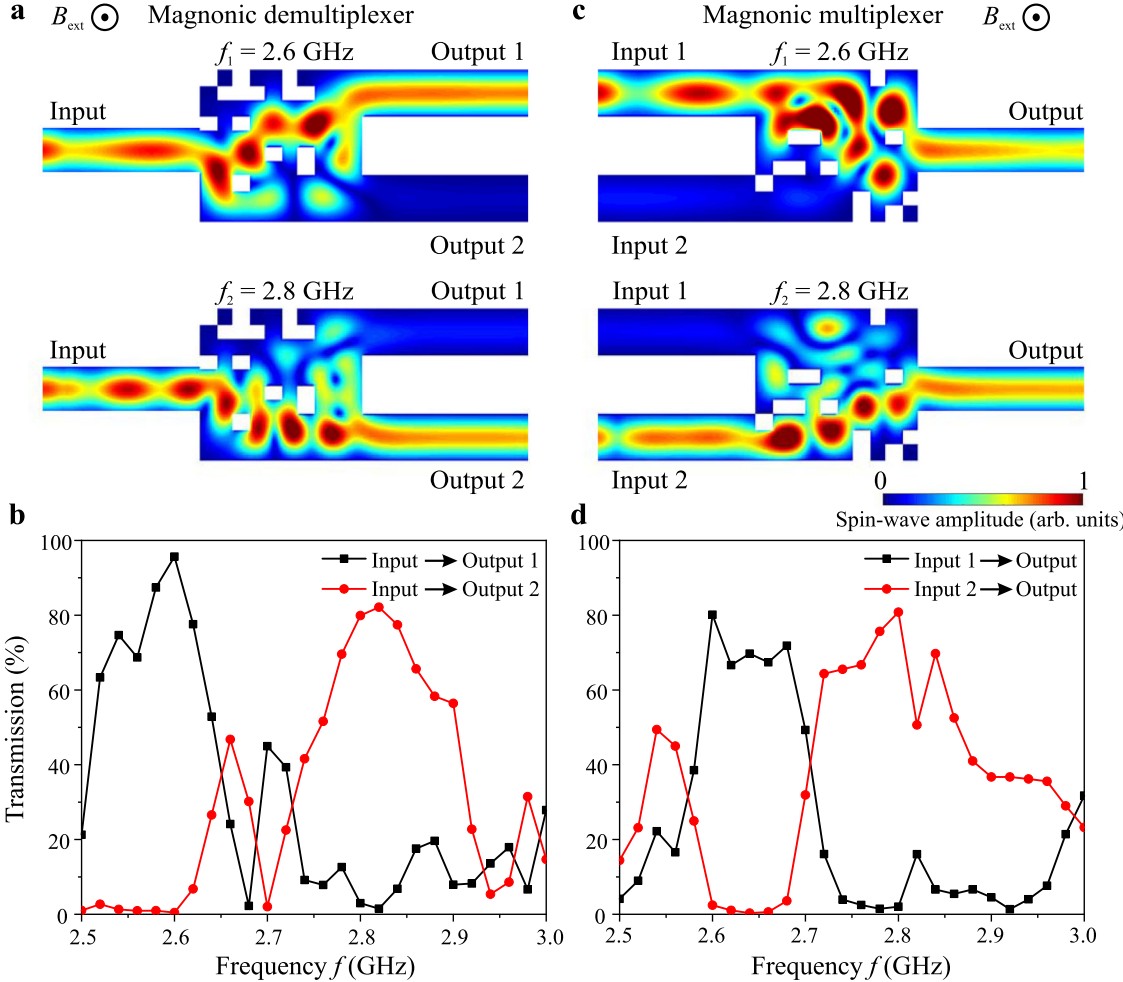

**Fig. 3 Working principle of inverse-designed magnonic demultiplexer and multiplexer. a** Demultiplexer: The normalized spin-wave amplitude map. The guiding of the waves of different frequency towards different outputs is clearly visible. **b** The simulated transmission of spin waves towards different output waveguides as a function of frequency simulated for the same optimized structure. A broad 3 dB bandwidth of the device around 120 MHz is found. **c, d** Multiplexer: the same dependencies for the magnonic multiplexer. Number of iterations is three.

of the powerful optimization algorithms (e.g., with the usage of deep machine learning approaches) will allow for the realization of the full inverse-design magnonics potential[46].

**Demonstrator 1: Magnonic frequency demultiplexer and multiplexer.** Figure. 3a shows the color maps of the normalized spin-wave amplitude in the designed device for different input frequencies. As expected, the spin wave of the frequency $f_1 = 2.6$ GHz is practically entirely guided into the first output waveguide, whereas the signal of the frequency $f_2 = 2.8$ GHz is almost fully guided into the second output waveguide. The crosstalk between the two frequencies is low (<3%). The functionality of the demultiplexer is mainly realized by the multi-path interference in the design region, i.e., the spin waves of frequency $f_1$ constructively interfere at output 1 and destructively interfere at output 2 and the opposite situation happens for frequency $f_2$ as shown in Fig. 3a (see Animation 1).

In order to analyze the frequency bandwidth of the designed device, the same structure is simulated by sweeping the frequencies from 2.5 GHz to 3 GHz. The transmission spectra at different output waveguides are shown in Fig. 3b. The transmission values are obtained by the normalizing of spin-wave intensity at the outputs with the intensity in a straight reference waveguide of the same length. In this way, the influence

of the spin-wave damping is excluded that allows us to study the insertion losses of the device. The transmission of frequencies $f_1$ and $f_2$ are around 96% and 80%, which correspond to the low insertion losses of 4% and 20%. These insertion losses are mainly caused by the reflections in the design region and at the input and output regions. As seen in Fig. 3b, the device exhibits a broad 3 dB bandwidth of around 120 MHz for both center frequencies, which implies that the design is robust with respect to the change in magnetic field or temperature shifting spin-wave dispersions. It is noted that this multiplexer is optimized for operations with the two frequencies of 2.6 GHz and 2.8 GHz only. The bandwidths can be further improved by using multifrequency optimization, in which several frequencies with equal spacing around each center frequency are chosen in the optimization procedure[27].

If the system is perfectly reciprocal, one can obtain the multiplexer just by reversing the spin-wave flow direction. However, as discussed below, a weak nonreciprocity exists in our system and the multiplexer based on the design shown in Fig. 3a is not optimal (see Supplementary information). So, we use the same algorithm design a dedicated multiplexer. It provides the inverse functionality of demultiplexer, i.e., combines two inputs signals into one output waveguide which is clearly shown in the Fig. 3c (see Animation 2). The objective function is defined as: $O = (I_{f_1,out} - I_{f_1,input_2})(I_{f_2,out} - I_{f_2,input_1})$, where $I_{f_i,out}$

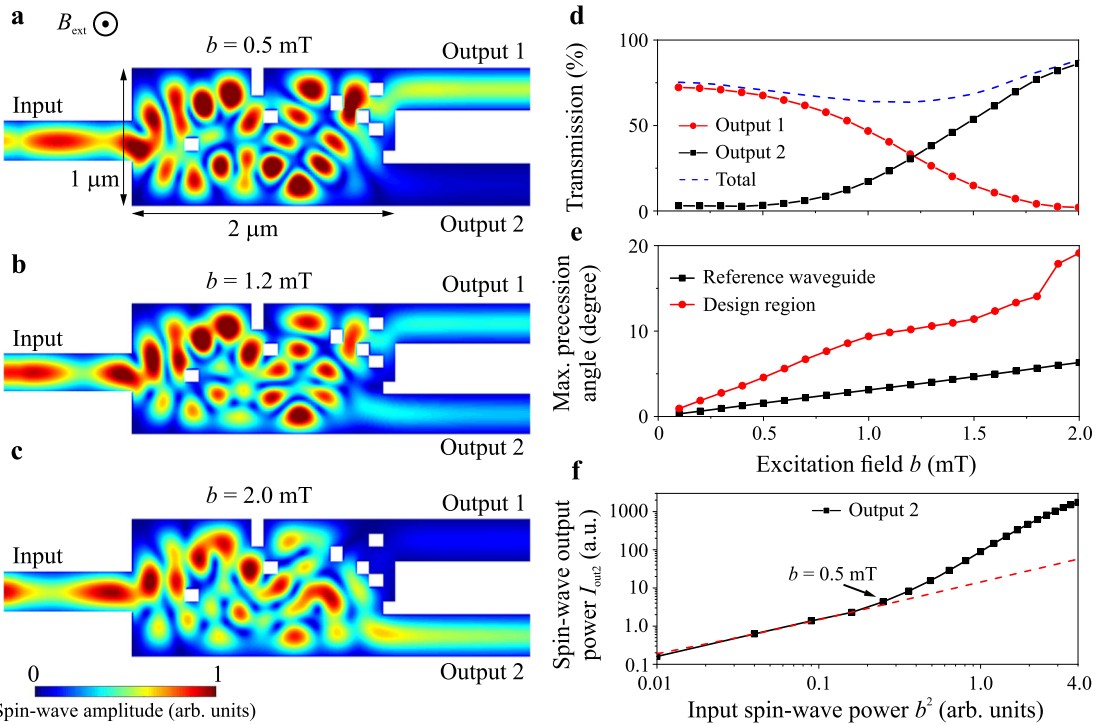

**Fig. 4 Inverse-design nonlinear magnonic switch. a–c** The normalized spin-wave amplitude distribution for excitation fields (**a**) $b = 0.5$ mT, (**b**) $b = 1.2$ mT, and (**c**) $b = 2.0$ mT. One can clearly see that the low-power spin wave is guided into output 1 and the high-power spin wave (the power is increased by a factor of 16 times) is guided to output 2. The design region was increased twice to reach high efficiency of the nonlinear switch. **d** The spin-wave transmission to the different output waveguides and the summarized total transmission (dashed blue line) as a function of the excitation field $b$. The decrease in the total transmitted power suggests that a part of spin-wave energy is stored in the form of standing waves in the structure. **e** The maximal precession angle in the reference straight waveguide (black dots line) and design region (red dots line) as a function of the excitation field $b$. **f** The absolute spin-wave output power at the second output waveguide as a function of the input power (the square of excitation field $b^2$) and the linear fit to the first four points (red dashed line). The nonlinear properties are getting pronounced for the microwave fields >0.5 mT. Number of iterations is three.

and $I_{f_i, input_j}$ are the spin-wave intensity of frequency $f_i$ at the output waveguide and at the $j$-th input waveguide ($i,j = 1,2$). This device also shows a high transmission of around 80% at $f_1$ and 81% at $f_2$ and a broad 3 dB bandwidth of around 120 MHz as shown in Fig. 3d. The influences of the simulation cell size, temperature, and the possible influence of inaccuracies in the nano-structuring process are shown in Supplementary Materials. Finally, the functionality of the miniaturized demultiplexer of the size of the design region of $100 \times 100$ nm$^2$ ($10 \times 10$ nm$^2$ element size) is presented in Supplementary Materials. The fundamental limitation of the element size coincides with the fundamental limitation of the spin-wave wavelength and is given by a lattice constant of the used magnetic material[3,47].

## Demonstrator 2: Nonlinear magnonic switch.

Nonlinear phenomena are required for data processing and pronounced nonlinear spin-wave properties are one of the significant added values offered by magnonics[10–13,23]. Here, we demonstrate the use of the inverse-design approach for the development of a nonlinear magnonic switch, in which a low-power spin wave is guided into one output waveguide while the high-power wave is guided into another one. Similar functionality was shown recently experimentally in the nanoscale directional coupler and was used for the development of a magnonic half-adder via the realization of AND and XOR logic gates[23]. Moreover, it allows for the realization of rf devices like power limiters or signal-to-noise enhancers, as well as, together with the nonlinear ring resonator[19], can be a key building block for future neuromorphic networks. To design the nonlinear switch, we keep the spin-wave frequency fixed at 2.6 GHz and run two parallel simulations with

two values of excitation fields ($b_1 = 0.5$ mT and $b_2 = 2$ mT) to operate with spin waves of different powers. To achieve the nonlinear functionality of high efficiency, the design region was increased twice to $1 \times 2$ μm$^2$ as it is shown in Fig. 4a. The objective function is defined as: $O = (I_{low,o_1} - I_{low,o_2})(I_{high,o_2} - I_{high,o_1})$, where $I_{low,o_i}$ and $I_{high,o_i}$ represent the spin-wave intensity for low and high input power at the $i$-th output waveguide ($i = 1,2$), respectively. Figure 4a–c clearly shows the switching between two outputs: (a) the low-power spin wave (@ $b = 0.5$ mT) is guided into the first output waveguide, (b) the medium-power spin wave (@ $b = 1.2$ mT) is guided into both output waveguides, (c) while the high-power wave (@ $b = 2$ mT) changes its path and is guided into the second waveguide (see Animation 3). The nonlinear switching can be explained by the power-dependent change in the effective saturation magnetization that shifts the dispersion curve[10,11,23]. For the case of FVSWs, the dispersion curve is shifted up with the increase in the magnetization precession angle[11] (see the Supplementary Information) what, consequently, results in the increase of the spin-wave wavelength at a fixed frequency. The change in the wavelength changes the interference pattern in the design region (see Fig. 4a-c and Animation 3) and, finally, switches the output channel.

Further, we investigate the continuous evolution of the functionality of the device in the range of fields $b$ from 0.1 mT to 2 mT. The spin-wave transmission towards the different outputs (red and black dots lines) and the total summarized transmitted spin-wave power (blue dashed line) are shown in Fig. 4d as a function of the excitation field. The total transmission is around 75% at $b = 0.1$ mT, and decreases down to 64% with the increase in the excitation field up to 1.2 mT. This indicates that either the total parasitic reflection is

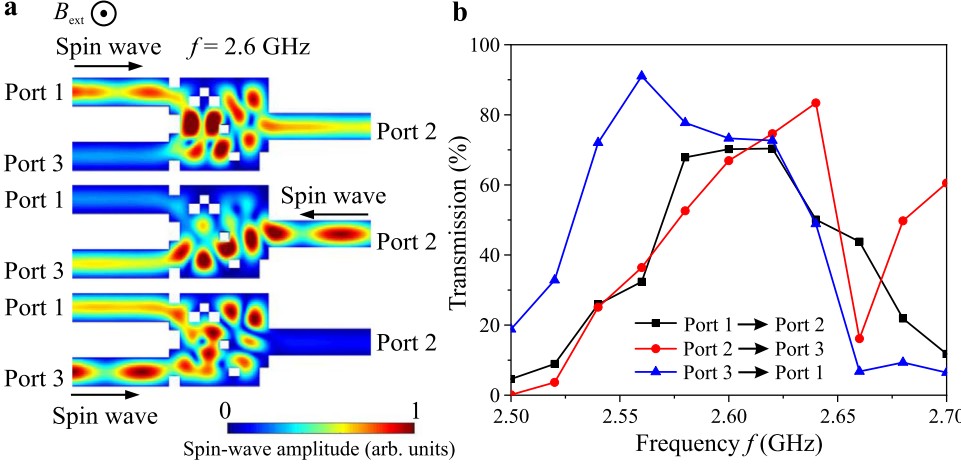

**Fig. 5 Nonreciprocal magnonic circulator. a** The normalized spin-wave amplitude distribution for different inputs at the same frequency of 2.6 GHz. Spin waves are excited at the three different ports and are guided to one of the other ports according to the rule: 1→2→3→1. **b** The transmissions of spin waves between different inputs (open direction) as a function of the frequency simulated for the same optimized device. A broad 3 dB bandwidth of around 100 MHz and a transmission for all the three cases close to 70% is observed (the crosstalk is smaller than 12%). The number of iterations is five.

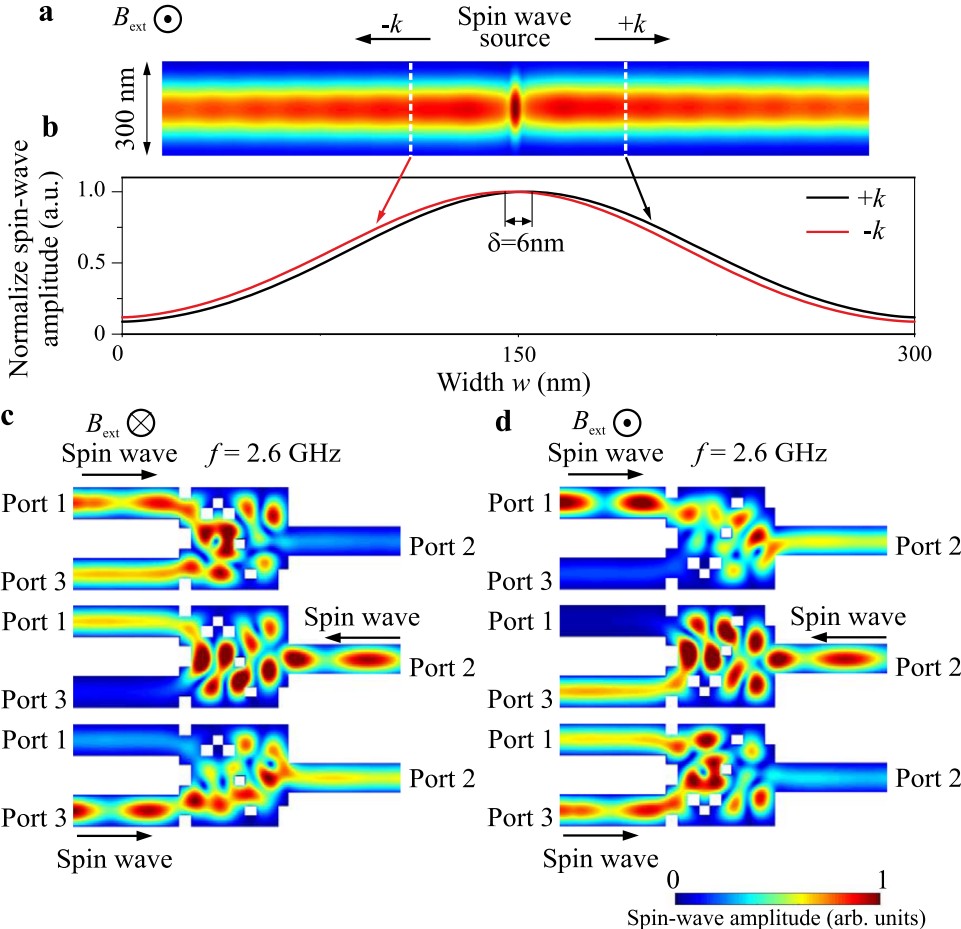

**Fig. 6 Physical phenomena behind the nonreciprocity of FVSWs. a** The normalized spin-wave amplitude in the single waveguide. Spin waves are excited in the middle of the waveguide and propagate in opposite directions. **b** The spin-wave profiles along the width of the waveguide for the positive and negative wavenumbers extracted from the regions as marked with the dashed white lines. The weak localization of the waves closer to one of the waveguide edges depending on the propagation directions is visible. **c,d** The normalized spin-wave amplitude in the same circulators with (**c**) inverted external field direction (the functionality 3→2→1→3 is achieved) and (**d**) with the original field direction like in Fig. 5a but with the flipped design structure along y-axis (the functionality 1→2→3→1 is achieved). The results show that nonreciprocal behavior is associated with the gyrotropic magnetization precession motion and breaking of time symmetry.

increased or the energy is temporary stored in the designed region in the form of standing waves[12,19]. Figure 4e shows the maximal precession angle in a straight reference waveguide (black dots line) and in the design region (red dots line) as a function of the excitation field. The increased precession angle in the design region clearly indicates the resonator-like behavior. A further increase in the excitation field up to 2 mT results in an increase of the total transmission up to 88%. The transmission increase can be explained by the change in the interference pattern that "releases" the stored energy–see the decreased spin-wave intensity inside the design region in Fig. 4c with respect to Fig .4a. The red and black dotted lines in Fig. 4d, clearly show the nonlinear switch functionality of the prototype device: The transmission at the second output waveguide is almost constant and stays below 3.3% for the excitation fields smaller than $b = 0.5$ mT (linear regime) but gradually increases up to 86% at the field of 2 mT due to the nonlinear shift. High isolation of around 26 times ($\sim$14 dB) between the two outputs is observed. Figure 4f shows the power of output 2 as a function of input spin-wave power ($b^2$) and the increase in the input power by a factor of 16 results in the output power increase by a factor of around 400 that can be used for signal-to-noise enhancement. The linear fit in the figure (red dashed line) also clearly shows that this nonlinearity starts to play a pronounced role when the external field exceeds the value of 0.5 mT ($b^2 = 0.25$).

**Demonstrator 3: Nonreciprocal magnonic circulator.** The realization of a nonreciprocal device, such as isolators and circulators, is of fundamental importance in electronic, photonic and magnonic systems. In the field of magnonics, this effect is usually achieved by the usage of ferromagnetic dipolar-coupled bilayer systems[21,22], interfacial Dzyaloshinskii–Moriya interaction[31,48] or complex topological structures[20]. However, these approaches are hard to be applied in real devices due to the comparably large damping in these systems. Here, we demonstrate the potential of inverse-design magnonics by a three-port magnonic circulator and the optimization algorithm "has found on its own" another mechanism to achieve nonreciprocal spin-wave properties. The mechanism is also based on the complex dynamic demagnetization tensors but, as opposite to the bilayers systems[21] or 3D structures[22], is achieved by the use of a nonintuitive 2D planar matrix made of the same material with low damping.

Similar to the previous optimization procedure, the spin wave frequency is fixed at 2.6 GHz and three parallel simulations are performed with the excitation of spin waves in different input waveguides. In order to realize the functionality of the magnonic circulator, the objective function of the optimization is defined as: $O = (I_{I_1, P_2} - I_{I_1, P_3})(I_{I_2, P_3} - I_{I_2, P_1})(I_{I_3, P_1} - I_{I_3, P_2})$, where $I_{I_i, P_j}$ represents the spin-wave intensity from the input $i$ to the port $j$ ($i,j = 1,2,3$). Figure 5a show the operation principle of the optimized magnonic circulator after five iterations: A spin-wave signal from port 1 is transmitted to port 2, the one from port 2 is transmitted to port 3, and finally, the third spin-wave signal from port 3 is transmitted to port 1 (see Animation 4). The same structure is used for additional simulations in the frequency range from 2.5 GHz to 2.7 GHz and the spectra for all three (open direction) cases are shown in Fig. 5b. The spectra show a broad 3 dB bandwidth of around 100 MHz and the transmission for all three cases is close to 70% at 2.6 GHz while the crosstalk is smaller than 12% (i.e., the isolation is larger than 7.7 dB). These characteristics can be further improved by the use of a larger matrix of elements or a wider parameters space as was discussed above. Moreover, the design itself (e.g., the positions of the inputs and outputs) can be optimized further since in this demonstrator we have used on purpose a design similar to the other devices.

In order to understand the physical phenomena staying behind the nonreciprocal functionality, additional simulations are performed in a single waveguide of the same width as the input and output waveguides in Fig. 5. It is noted that the FVMSWs are well known to be isotropic and reciprocal in an unstructured film[49]. Figure 6a shows the normalized amplitude of a spin wave excited at the center of the waveguide (waves propagating to the opposite directions marked as $-k$ and $+k$). Figure. 6b shows the spin-wave profiles across the width of the waveguide extracted from the regions marked with dashed white lines. A small shift around 6 nm is clearly observed in the profiles and indicates that the FVSWs in the nanoscale waveguide are nonreciprocal in a way typical for Damon–Eshbach geometry, in which the external field is applied in-plane transversally to the waveguide[15,49,50].

This nonreciprocity is caused by the off-diagonal element of the dipole–dipole interaction, which breaks time-reversal symmetry[49,50]. Our case can be considered as a 90-degree rotated Demon–Eshbach geometry and the spin waves are localized at one of the waveguide edges (depending on the direction of propagation) rather than on the surfaces. It is noted that the nonreciprocity observed here is much weaker than in the classical case of Demon–Eshbach geometry in a thick film since also the exchange interaction defines the properties of spin waves at the nanoscale in addition to the dipolar contribution[30,50]. To double check the statement on the nonreciprocity, the same simulations are performed with the reversed external field direction $B_{ext} = -200$ mT–see Fig. 6c. As expected, the circulation of the magnonic circulator is inverted from $1{\rightarrow}2{\rightarrow}3{\rightarrow}1$ (Figs. 5a) to $3{\rightarrow}2{\rightarrow}1{\rightarrow}3$ (Fig. 6c) due to the reverse of the magnetization precession from counterclockwise to clockwise (top view). Furthermore, additional simulations are performed with the design region (matrix of elements) to be flipped along the $y$-axis as shown in Figure 6d. The obtained result is similar to the original one shown in Figure 5a, which indicates that the nonreciprocity mainly depends on the direction of the external field, i.e., the direction of the magnetization precession. Additional simulations with increased damping and two cells along the thickness of the structure are shown in the Supplementary Information to exclude other possible mechanisms behind the nonreciprocity. Finally, we would like to emphasize that the functionality of the magnonic circulator is realized by the accumulation of the rather weak intrinsic nonreciprocity of FVSWs in a nanoscale structure. This suggests that the inverse-design approach can be also used for the finding and exploration of the up-to-know overlooked potential capacity of physical phenomena for applications.

## Discussion

We have developed an inverse-design method for magnonics and demonstrated its capabilities and potential by designing various linear, nonlinear, and nonreciprocal magnonic devices. The simulated (de-)multiplexer, nonlinear switch, and circulator have similar architectures and their functionalities were achieved automatically by the feed-back DBS optimization algorithm. The proposed devices have compact footprints (down to $200 \times 100$ nm$^2$), exhibit high efficiency with high potential for further improvement, and can be easily manufactured as well as scaled down to the CMOS scale. We expect that any other functionality of the device for microwave application, Boolean logic or unconventional computing can be achieved using the same approach because particularly inverse-design magnonics allows for the simultaneous operation with nonlinear and nonreciprocal waves. Moreover, the inverse-design algorithm attracted our attention to the week nonreciprocity of Forward Volume Spin Waves in the nanostructures and accumulated it for the realization of the magnonic circulator. We sincerely believe that inverse-design magnonics provides a new significant momentum to the development of the applied solid-state physics and technology.

## Methods

**Micromagnetic simulations.** The micromagnetic simulations were performed by the GPU-accelerated simulation package Mumax3, including both exchange and dipolar interactions, to calculate the space- and time-dependent magnetization dynamics in the investigated structures[44]. The parameters of a nanometer thick YIG film were used:[33] saturation magnetization $M_s = 1.4 \times 10^5$ A/m, exchange constant $A = 3.5$ pJ/m, and Gilbert damping $\alpha = 2 \times 10^{-4}$. In our simulations, the Gilbert damping at the end of the device was set to exponentially increase to 0.5 to avoid spin-wave reflection. The mesh was set to $20 \times 20$ nm$^2$. A smaller mesh size ($10 \times 10$ nm$^2$) was also checked in Supplementary Information and did not show a significant difference. An external field $B_{ext} = 200$ mT is applied along the out-of-plane axis ($z$-axis as shown in Fig. 1) which is sufficient to saturate the structure along this direction. To excite a propagating spin wave, we applied a sinusoidal magnetic field $b_y = b\sin(2\pi ft)$ at the input waveguide over an area of 300 nm in length, with a varying microwave frequency of $f$. The operational characteristics in the case of the applied sinc pulse is discussed in Supplementary Materials. If it is not stated otherwise in the paper, the amplitude of the excitation field $b$ is 0.1 mT. The $M_y(x,y,t)$ of each cell was collected over a period of 50 ns and recorded in 50 ps intervals during the optimization procedure. This time is increased to 100 ns for the simulations of frequency (power) sweeping which is long enough to reach a stable dynamic equilibrium. The fluctuations $m_y(x,y,t)$ were calculated for all cells via $m_y(x,y,t) = M_y(x,y,t) - M_y(x,y,0)$, where $M_y(x,y,0)$ corresponds to the ground state. The spin-wave intensities were calculated performing a fast Fourier transformation for a dynamic equilibrium range. If it is not mentioned, all simulations were performed without taking temperature into account. The influence of temperature is also investigated in the Supplementary Information. The whole size of the simulated structure in our optimization procedure was $10240 \times 1280$ nm$^2$ including $512 \times 64$ cells. The time for one single simulation is around 45 s with two gaming graphic cards (TITAN xp GPUs). One iteration (including 100 simulations for the inverse-design of the magnonic (de-)multiplexer and circulator) took roughly 1.3 h, in total, it took about 4–7 h to optimize one device. The total optimization time depends on the mesh size, the whole structure size, design region size and the convergence speed.

## Data availability

The data that support the plots presented in this paper are available from the corresponding authors upon reasonable request.

## Code availability

The code used to analyze the data and the related simulation files are available from the corresponding author upon reasonable request.

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

## Acknowledgements

The authors thank Adam Papp, Gyorgy Csaba, Markus Becherer, Claas Abert and Dieter Süss for the valuable discussions. The project is funded by the European Research Council (ERC) Starting Grant 678309 MagnonCircuits, the Austrian Science Fund (FWF) for support through Grant No. I 4917-N (MagFunc) and the Deutsche Forschungsgemeinschaft (DFG, German Research Foundation) - TRR 173 - 268565370 ("Spin + X", Project B01).

## Author contributions

Q. W. conceived the idea and carried out the micromagnetic simulations. P. P. and A. V. C led this project. Q. W. wrote the manuscript with the help of all the coauthors. All authors contributed to the scientific discussion and commented on the manuscript.

## Competing interests

The authors declare no competing interests.
