## [Peer Review File · Nature Communications]

Reviewers' Comments:

Reviewer #1:

Remarks to the Author:

This paper proposes to construct generic magnonic devices based on inverse design technique, which has been applied in constructing optical devices. The characters of magnonic devices are very similar to that of optical devices. In comparison with the negligible of non-linear effect in optical systems, the non-linear effect in magnonic systems is more prominent and ready to be utilized in device applications. This paper first demonstrates the realization of magnonic frequency demultiplexer and multiplexer, which are purely linear devices based on interference only. Then the paper shows two non-linear magnonic devices: the non-linear magnonic switch and the non-reciprocal magnonic circulator. The numerical method of mumax in combination with the direct binary search (DBS) algorithm is standard and reliable for the study of the paper. The details provided in the paper are sufficient in reproducing the described results.

This paper demonstrates the application of inverse-design technique on designing magnonic devices, and utilizing the non-linear effect in magnons to realize non-linear devices. This work would inspire people to investigate further the potential capabilities of the inverse-design method in magnetics, how would this method generate more useful magnonic device designs. Therefore, I would like to recommend the paper to be published with the following questions for the authors to comment on.

- 1) This paper gives a numerical simulation of constructing magnonic device via inverse-design method. To avoid confusion, it should be pointed out clearly in the title or in the abstract that it is 'numerical simulation', not experimental realization of such devices.
- 2) In the mumax simulation, has the dipolar interaction been included in all simulations? If not, how would dipole fields influence the device behavior.
- 3) Can the non-linear magnonic switch and the demultiplexer be regarded as the same device? Because the low input power spin wave has higher frequency than the high power spin wave, so a demultiplexer that differentiate frequencies can also differentiate power.
- 4) The inverse-design method could be quite useful in designing a standalone device. However, to scale up, multiple devices needs to be connected. How good would this method in giving solutions in scaled magnonic circuits?

Reviewer #2:

Remarks to the Author:

Authors of the manuscript implement an inverse design algorithm for magnonic devices. To my understanding it is, together with another manuscript simultaneously under review, a first implementation of this algorithm in the field of magnonics. The authors demonstrate that this method is very powerful and can lead to design of functional magnonic elements with simple geometries. I believe that the experimental works based on this work will follow. In addition, the authors exploit some effects that are characteristic for spin waves, nonlinearity and nonreciprocity. They demonstrate that very small nonreciprocal effect is present in forward volume geometry and explain its origin. Further they demonstrate that the small nonreciprocal effect is sufficient to have impact on the functionality of designed elements, reverse demultiplexer, or circulator. Overall I consider that the results are presented in details and in clear way. I support the publication of the manuscript in Nature Communications, without any changes.

Authors can consider suggestion:

"In order to analyze the frequency bandwidth of the designed device, the same structure is simulated by sweeping the frequencies from 2.5 GHz to 3 GHz."

According to Fig 3 b d, the Authors used multiple simulations for each a single sinusoidal magnetic field was used to excite the SWs. The step function was 0.1 GHz, which I agree it is sufficient to demonstrate the characteristic of the demultiplexer presented here. Alternatively, the excitation could be realized in one simulations with sinc function and fourier transform could be avaluated at the ports. This function should excite the sine waves with equal power input. Therefore the Fig 3 b d could be calculate faster with larger precision.

Reply to Referee #1:

The Reviewer #1 wrote:

This paper proposes to construct generic magnonic devices based on inverse design technique, which has been applied in constructing optical devices. The characters of magnonic devices are very similar to that of optical devices. In comparison with the negligible of non-linear effect in optical systems, the non-linear effect in magnonic systems is more prominent and ready to be utilized in device applications. This paper first demonstrates the realization of magnonic frequency demultiplexer and multiplexer, which are purely linear devices based on interference only. Then the paper shows two non-linear magnonic devices: the non-linear magnonic switch and the non-reciprocal magnonic circulator. The numerical method of mumax in combination with the direct binary search (DBS) algorithm is standard and reliable for the study of the paper. The details provided in the paper are sufficient in reproducing the described results.

This paper demonstrates the application of inverse-design technique on designing magnonic devices, and utilizing the non-linear effect in magnons to realize non-linear devices. This work would inspire people to investigate further the potential capabilities of the inverse-design method in magnetics, how would this method generate more useful magnonic device designs. Therefore, I would like to recommend the paper to be published with the following questions for the authors to comment on.

We thank the Referee for the high evaluation of our results.

1. *This paper gives a numerical simulation of constructing magnonic device via inverse-design method. To avoid confusion, it should be pointed out clearly in the title or in the abstract that it is 'numerical simulation', not experimental realization of such devices.*

We would like to thank the Referee for making this point. In the revised version, we have added the following information (red part) into the abstract “**Here, we introduce the method of inverse-design magnonics, in which any functionality can be specified first, and a feedback-based computational algorithm is used to obtain the device design. We validate this method using the means of micromagnetic simulations.**”

2. *In the mumax simulation, has the dipolar interaction been included in all simulations? If not, how would dipole fields influence the device behavior.*

Yes, the dipolar interaction is included in all our simulations. The weak nonreciprocity is caused by the off-diagonal element of the dipole-dipole interaction. In order to clarify this point, we have added the information into the Methods. “**The micromagnetic simulations were performed by the GPU-accelerated simulation package Mumax³, including both exchange and dipolar interactions, to calculate the space- and time-dependent magnetization dynamics in the investigated structures [44].**”

3. *Can the non-linear magnonic switch and the demultiplexer be regarded as the same device? Because the low input power spin wave has higher frequency than the high power spin wave, so a demultiplexer that differentiate frequencies can also differentiate power.*

Yes, these two devices are similar. The demultiplexer is used to separate the spin waves with different frequencies, and the nonlinear magnonic switch could also be used to separate the spin waves with different input powers. The functionalities of these devices are mainly realized by the multi-path interference in the design region. For the frequency demultiplexer, the situation is intuitively clear, i.e., the spin wave of frequency f_1 (wavelength λ_1) constructively interfere at output 1 and destructively interfere at output 2 and the opposite situation happens for frequency f_2 (wavelength λ_2) as shown in Fig. 3a. For the nonlinear magnonic switch, the frequency of all the spin-wave input power is fixed at 2.6 GHz. In this regime of moderate nonlinearity, the frequency does not change when we increase the input power. Thus, we are *not* dealing with the situation of magnonic instabilities that would create other spin-wave frequencies in addition to the input frequency. However, the spin-wave wavelength will decrease due to the nonlinear shift (see Supplementary Materials Section 6). Because of the different wavelengths of the spin wave at different powers, the interference patterns are different. As a result, the spin waves with different powers are guided into different outputs.

4. *The inverse-design method could be quite useful in designing a standalone device. However, to scale up, multiple devices needs to be connected. How good would this method in giving solutions in scaled magnonic circuits?*

We thank the Referee for raising this discussion. We think there are few ways to design a complex circuit.

1. The most straightforward way is to design individual magnonic devices using the inverse-design method and then connect them. To achieve this, additional optimizations are needed during the procession of the connection. In addition, spin-wave amplifiers might need to be installed in between the individual units [Nat. Electron. 3, 765 (2020)].
2. The whole magnonic circuit is directly designed using the inverse-design method. For this case, the circuit should be more compact and additional optimizations are not needed. However, the DBS is not the best way for large design regions since the simulation time will dramatically increase with the increase in the size of the design region. Therefore, the investigations have to be focused on the further optimization of numeric simulations (e.g., an eigenmodes solver could make simulations much faster) and on the search of new algorithms (e.g., the gradient-based algorithm associated with deep machine learning) which will make inverse-design magnonics more efficient. However, we expect that despite potential improvements of the method in general, large computing power will always be a cornerstone of inverse design of large circuits.
3. Hybrid system: Most of the devices (like directional couplers, ring resonators ...) in the circuits are still taken from the classical design. The inverse-design method is employed to locally modified a part of the design to realize a special functionality. Nowadays, researchers in photonics are using this way to design large and complex photonic circuits [Nat. Photon. 14, 369 (2020)].

Reply to Referee #2:

The Reviewer #2 wrote:

Authors of the manuscript implement an inverse design algorithm for magnonic devices. To my understanding it is, together with another manuscript simultaneously under review, a first implementation of this algorithm in the field of magnonics. The authors demonstrate that this method is very powerful and can lead to design of functional magnonic elements with simple geometries. I believe that the experimental works based on this work will follow. In addition, the authors exploit some effects that are characteristic for spin waves, nonlinearity and nonreciprocity. They demonstrate that very small nonreciprocal effect is present in forward volume geometry and explain its origin. Further they demonstrate that the small nonreciprocal effect is sufficient to have impact on the functionality of designed elements, reverse demultiplexer, or circulator. Overall I consider that the results are presented in details and in clear way. I support the publication of the manuscript in Nature Communications, without any changes.

We would like to thank the Referee for his/her high evaluation of our results.

Authors can consider suggestion:

"In order to analyze the frequency bandwidth of the designed device, the same structure is simulated by sweeping the frequencies from 2.5 GHz to 3 GHz."

According to Fig 3 b d, the Authors used multiple simulations for each a single sinusoidal magnetic field was used to excite the SWs. The step function was 0.1 GHz, which I agree it is sufficient to demonstrate the characteristic of the demultiplexer presented here. Alternatively, the excitation could be realized in one simulations with sinc function and fourier transform could be evaluated at the ports. This function should excite the sine waves with equal power input. Therefore the Fig 3 b d could be calculate faster with larger precision.

We appreciate the Referee's question. The suggested method is a powerful way to calculate spin-wave spectra or dispersion curves. We also used the same way to calculate the spin-wave spectra/dispersion curves in single waveguides or magnonic crystals in our previous works [Sci. Adv. 4, e1701517 (2018) and Phys. Rev. B 95, 134433 (2017)]. Nevertheless, we should emphasize that in the case of the sinc pulse, the device operates in a pulse regime (namely it operates with many waves of different frequencies and nonstationary amplitudes simultaneously) as opposed to the originally investigated "single-frequency" continuous-wave (CW) regime. Thus, the comparison between the pulse and the CW regimes represents an interesting task.

Following the Referee's suggestion, we performed the simulation for the operation of the magnonic demultiplexer in the pulse regime and obtained spin-wave spectra at different outputs (see Fig. R1). The result is very similar to the one presented in the original manuscript (Fig. 3b). This points on the applicability of the approach proposed by the Referee. However, the minimum transmission power is a little bit higher (~3% at 2.6 GHz, ~19% at 2.8 GHz) than what we show in the manuscript in the CW regime (~0.5% at 2.6 GHz, ~3% at 2.8 GHz). The small difference can be explained by the following reason: The functionality of the demultiplexer is mainly

provided by the wave interference, and the device is optimized for the CW regime when all waves have constant amplitudes. Thus, if to apply a CW signal to the device, a “transition region” exists before the system reaches a stable interference pattern. This “transition region” mimics the pulsed operational regime of the device.

Figure R1. The spin-wave power of different output waveguides as a function of frequency simulated using sinc function excitation.

Figure R2 shows the dynamic magnetization (M_z) as a function of simulation time. Three distinct regions are marked in the figure: delay time region, transition region and the region of dynamic equilibrium. In our manuscript, we only perform the FFT in the dynamic equilibrium region which excludes the contribution of the transition region. However, the contribution of the transition region is included in the method of the sinc function excitation and cannot be separated. Nevertheless, the method suggested by the Referee is much faster and could be used for the estimation of the operating frequencies of the designed devices. An optimized method would therefore start with a sinc function to obtain a first characterization of the spectra at the outputs and to optimize the frequency separation. Further improvement can then be obtained by testing the relevant frequencies in the dynamic equilibrium using sinusoidal excitation. Another point is that the delay time and the duration of the transition region depend on the group velocity of the spin waves and the size of the devices. It can be further decreased by using high-speed exchange waves or decreasing the size of the devices. Therefore, we expect that the difference between the two methods will reduce further when the size of the device is scaled down and ultrafast exchange spin waves are applied. Finally, we believe that a separate optimization of the device for the operations with short (most likely, rectangular) pulses is also possible using the presented optimization algorithm. But we would like to leave this topic beyond the scope of the current manuscript for the consistency. However, the information about the operation of the device in the case of applied sinc pulse, which answers already many questions, is added to the Supplementary Materials.

Figure R2. The dynamic magnetization (M_z) as a function of simulation time at different outputs of demultiplexer for operation frequency of 2.6 GHz. Three different regions marked.

Reviewers' Comments:

Reviewer #1:

Remarks to the Author:

The authors answered all questions from both reviewers in previous reports in a satisfactory manner, therefore I support the publication as is.

Reply to Referee #1:

The Reviewer #1 wrote:

The authors answered all questions from both reviewers in previous reports in a satisfactory manner, therefore I support the publication as is.

We thank the Referee for the positive evaluation. No modification of the manuscript was required.